# MnTBAP Reverses Pulmonary Vascular Remodeling and Improves Cardiac Function in Experimentally Induced Pulmonary Arterial Hypertension

**DOI:** 10.3390/ijms21114130

**Published:** 2020-06-10

**Authors:** Maria Catalina Gomez-Puerto, Xiao-Qing Sun, Ingrid Schalij, Mar Orriols, Xiaoke Pan, Robert Szulcek, Marie-José Goumans, Harm-Jan Bogaard, Qian Zhou, Peter ten Dijke

**Affiliations:** 1Department of Cell and Chemical Biology and Oncode Institute, Leiden University Medical Center, 2300 RC Leiden, The Netherlands; M.C.Gomez_Puerto@lumc.nl (M.C.G.-P.); marorriols@gmail.com (M.O.); M.J.T.H.Goumans@lumc.nl (M.-J.G.); P.ten_Dijke@lumc.nl (P.t.D.); 2Department of Pulmonary Medicine, Amsterdam Cardiovascular Sciences, Amsterdam UMC, Vrije Universiteit Amsterdam, 1081 HZ Amsterdam, The Netherlands; x.sun@amsterdamumc.nl (X.-Q.S.); i.schalij@amsterdamumc.nl (I.S.); x.pan@amsterdamumc.nl (X.P.); r.szulcek@amsterdamumc.nl (R.S.); 3Department of Cardiology and Angiology I, Heart Center Freiburg University, Faculty of Medicine, University of Freiburg, 79106 Freiburg, Germany; qian.zhou@usb.ch

**Keywords:** autophagy, BMPR2, MnTBAP, pulmonary arterial hypertension (PAH), human pulmonary arterial endothelial cells (PAECs), inflammation

## Abstract

Pulmonary arterial hypertension (PAH) is a life-threatening disease characterized by obstructed pulmonary vasculatures. Current therapies for PAH are limited and only alleviate symptoms. Reduced levels of BMPR2 are associated with PAH pathophysiology. Moreover, reactive oxygen species, inflammation and autophagy have been shown to be hallmarks in PAH. We previously demonstrated that MnTBAP, a synthetic metalloporphyrin with antioxidant and anti-inflammatory activity, inhibits the turn-over of BMPR2 in human umbilical vein endothelial cells. Therefore, we hypothesized that MnTBAP might be used to treat PAH. Human pulmonary artery endothelial cells (PAECs), as well as pulmonary microvascular endothelial (MVECs) and smooth muscle cells (MVSMCs) from PAH patients, were treated with MnTBAP. In vivo, either saline or MnTBAP was given to PAH rats induced by Sugen 5416 and hypoxia (SuHx). On PAECs, MnTBAP was found to increase BMPR2 protein levels by blocking autophagy. Moreover, MnTBAP increased BMPR2 levels in pulmonary MVECs and MVSMCs isolated from PAH patients. In SuHx rats, MnTBAP reduced right ventricular (RV) afterload by reversing pulmonary vascular remodeling, including both intima and media layers. Furthermore, MnTBAP improved RV function and reversed RV dilation in SuHx rats. Taken together, these data highlight the importance of MnTBAP as a potential therapeutic treatment for PAH.

## 1. Introduction

Pulmonary arterial hypertension (PAH) is a life-threatening disease characterized by obstructed pulmonary vasculature due to deregulated proliferation, migration and survival of pulmonary vascular cells (i.e., smooth muscle cells (SMCs) and endothelial cells (ECs)) for which the underlying mechanisms are not well understood [1]. Patients suffering from PAH have an increase in mean pulmonary arterial pressure leading to right ventricular (RV) failure [2]. Currently, no curative treatment is available and despite the efforts therapeutic options for PAH are limited and only alleviate symptoms [3,4,5]. Although rare, the disease occurs most frequently at a young age often even in childhood [6], and females have a three-fold increase in developing PAH [7]. Upon diagnosis, the average life expectancy of PAH patients is approximately 7–10 years and only lung transplantation offers any survival potential, which is not a long term solution as treatment [8].

There are different etiologies for PAH, including hereditary PAH (hPAH) and idiopathic PAH (iPAH). While both forms are clinically indistinguishable, the underlying etiology is different. hPAH is associated with inheritable genetic defects. Heterozygous germ-line mutations in *BMPR2*, a gene encoding the bone morphogenetic protein (BMP) type 2 receptor, are the most common causal factors in hPAH [9]. Interestingly, in both hPAH and iPAH, a reduction in BMPR2 expression in ECs has been reported [10,11,12]. Besides, since not all individuals carrying mutations in *BMPR2* will develop PAH, environmental factors including hypoxia and inflammation may provide local triggers for the disease [13,14,15,16,17].

Rescuing BMPR2 expression, function or signaling represents a promising treatment for PAH patients [18,19,20]. Manganese (III) tetrakis (4-benzoic acid) porphyrin (MnTBAP), a synthetic metalloporphyrin with antioxidant [21,22,23] and anti-inflammatory [23,24,25,26] effects, has been shown to inhibit the turn-over of BMPR2 in human umbilical vein endothelial cells (HUVECs) [25]. Moreover, MnTBAP offers beneficial effects in bleomycin-induced pulmonary fibrosis [27], carrageenan-induced pleurisy [28], lung contusion [26], renal fibrosis [29] and renal injury [24,30].

We and others have reported that endogenous BMPR2 is degraded through the lysosome in primary human pulmonary artery endothelial (PAECs) and smooth muscle cells (PASMCs) and that autophagy activation contributes to BMPR2 degradation [12,19,31,32]. In the present study, we show that partly by blocking autophagy, MnTBAP increases BMPR2 levels in pulmonary microvascular endothelial cells (MVECs) isolated from iPAH patients. Furthermore, for the first time, we demonstrate that MnTBAP reverses experimental PAH and improves cardiac function. Taken together, these data highlight the importance of MnTBAP as a potential therapeutic treatment for PAH.

## 2. Results

### 2.1. MnTBAP Increases BMPR2 Levels In Vitro by Inhibiting Autophagy

To investigate whether MnTBAP treatment increases BMPR2 protein levels in the context of PAH, primary human PAECs were treated with MnTBAP and the lysosomal inhibitor bafilomycin A1 (BafA1) as a positive control. As expected, BMPR2 levels were significantly increased after BafA1 treatment [12] (Figure 1A). Consistent with our previous findings [25], MnTBAP treatment resulted in a dose-depndent increase of BMPR2 protein levels in PAECs (Figure 1A or B). No significant differences on *BMPR2* mRNA levels were observed after MnTBAP treatment, indicating no changes at the transcriptional level (Figure 1C).

In addition, PAECs treated with MnTBAP in the presence of the protein synthesis inhibitor cycloheximide (CHX) show an increase in BMPR2, suggesting that MnTBAP mechanism of action does not depend on protein translation (Figure 2A). Since BMPR2 is degraded through the lysosomal pathway in an autophagy related fashion, we investigated whether MnTBAP could modulate autophagy. The levels of the autophagy markers microtubule associated protein 1 light chain 3 beta-II (MAP1LC3B-II) and sequestosome 1 (SQSTM1) were measured by western blotting analysis. Interestingly, both MAP1LC3B-II and SQSTM1 protein levels augmented after MnTBAP treatment (Figure 2B). An increase in MAP1LC3B-II or SQSTM1 could be interpreted as an increase in autophagic flux or as a decrease in functional autophagy due to a block in autophagosome degradation. To elucidate whether changes in MAP1LC3B-II and SQSTM1 correspond to an increase or a block in autophagic flux, PAECs were treated with MnTBAP in combination with BafA1 to prevent lysosomal degradation and block the fusion of autophagosomes with lysosomes. PAECs treated with MnTBAP alone show similar MAP1LC3B-II and SQSTM1 levels when compared to cells treated with MnTBAP and BafA1 (Figure 2B). To validate our findings, autophagy changes were measured by flow cytometry using Cyto-ID, a dye selectively labelling autophagic vacuoles (Figure 2C). PAECs treated with MnTBAP alone or MnTBAP in combination with hydroxychloroquine (HCQ) (an inhibitor of lysosomal function acting in a comparable way to BafA1) showed an equal accumulation of autophagic vacuoles (Figure 2C). Taken together, these results indicate that MnTBAP impairs functional autophagy in vitro.

### 2.2. MnTBAP Increases the Levels of BMPR2 in Pulmonary MVECs and MVSMCs Isolated from iPAH Patients

Besides heterozygous mutations in the *BMPR2* gene, inflammation has been identified as a crucial factor in the pathogenesis of PAH [13,15]. In particular, the pro-inflammatory cytokine tumor necrosis factor alpha (TNF-α) has been shown to reduce BMPR2 transcription in PAECs and PASMCs [13]. To elucidate whether MnTBAP could increase BMPR2 levels in the presence of inflammation, PAECs were treated with TNF-α and MnTBAP. As expected, a decrease in BMPR2 protein levels was observed after TNF-α treatment (Figure 3A). Interestingly, in the presence of TNF-α and MnTBAP, BMPR2 protein levels were rescued when compared to TNF-α alone (Figure 3A). To further investigate whether MnTBAP could be a potential treatment for PAH, MVECs isolated from iPAH patients were treated with the compound. As reported before [12], the upper band of BMPR2, which corresponds to the fully glycosylated mature form of the receptor [33,34], was not observed in any iPAH MVECs compared with control MVECs (Figure 3B). Importantly, an increase in BMPR2 levels after MnTBAP treatment was detected in both control MVECs and iPAH MVECs (Figure 3B). Consistent changes in pSMAD1/5 protein levels were not observed after MnTBAP treatment.

The close interaction between SMCs and ECs is important for vessel formation and maintenance, and was shown to be involved in PAH pathogenesis [35,36]. Therefore, we investigated whether MnTBAP could also upregulate BMPR2 levels in microvascular MVSMCs from iPAH patients. An increase in BMPR2 levels was observed in MnTBAP treated iPAH MVSMCs by western blotting analysis (Figure 3C). A block in autophagy after MnTBAP treatment was confirmed by measuring MAP1LC3B-II through western blotting analysis (Figure 3C).

### 2.3. MnTBAP Reduced RV Afterload in Experimental PAH

To assess the therapeutic effect of MnTBAP in vivo, we treated SuHx rats with MnTBAP from week 6 to week 10 after induction of PAH. No significant difference in survival was found between the vehicle group and MnTBAP treated group (Figure 4A). Rats exposed to the SuHx protocol showed signs of severe PAH at week 6 as confirmed by echocardiography [37]. As shown by pressure–volume analysis, 4 weeks treatment with MnTBAP significantly reduced RV afterload in SuHx-induced PAH rats, as revealed by reduced arterial elastance (Ea) at week 10 (Figure 4B). Consistently, the echocardiography analysis revealed that total pulmonary resistance (TPR) was significantly reduced by MnTBAP treatment from week 6 to week 10 (Figure 4C). Meanwhile, right ventricular systolic pressure (RVSP) as well as pulmonary artery acceleration time normalized to cycle length (PAAT/cl%) remained unchanged by the treatment (Figure 4B,C).

Moreover, MnTBAP treatment reduced RV diastolic stiffness at week 10 as revealed by reduced end diastolic elastance (Eed) (Figure 4D), while RV contractility and RV-arterial coupling were unaffected as shown by end systolic elastance (Ees) and Ees/Ea, respectively (Figure 4E). Accordingly, the echocardiography analysis shows that MnTBAP treatment improved stroke volume (SV) and tricuspid annular plane systolic excursion (TAPSE) from week 6 to week 10 (Figure 4F). In addition, MnTBAP partly reversed RV remodeling as shown by reduced RV end diastolic diameter (RVEDD) (Figure 4G).

To further elucidate the origin of the reduced RV afterload, we performed histology on the lungs to measure pulmonary vascular remodeling (Figure 5A). MnTBAP treatment reversed pulmonary vascular remodeling, as shown by a significantly reduced number of occluded vessels after the treatment (Figure 5B). Representative images of open, partly remodelled and an occluded vessel are shown in Appendix A. Further quantification of remodeling showed that MnTBAP reduced intima remodeling in all pulmonary vessels sized up to 100 μm (Figure 5C), and reduced media remodeling in pulmonary vessels between 60–100 μm (Figure 5D). Details on pulmonary vascular remodeling quantification can be found in Appendix A.

Since MnTBAP is a synthetic metalloporphyrin with antioxidant effect, we examined the effect of MnTBAP on oxidative stress. Immunofluorescence staining with anti-8-Oxo-2′-deoxyguanosine (8-OHdG) revealed that MnTBAP treatment reduced oxidative stress in the pulmonary vasculature (Figure 5E,F).

The effect of MnTBAP on RV afterload and vascular remodeling was related to a block in autophagy (Figure 1 and Figure 3C). Vessels from MnTBAP-treated SuHx rats showed a trend of reduced MAP1LC3B expression (Appendix A). Quantification was done including both ECs and SMCs. However, western blots from lung homogenates did not show any changes in MAP1LC3B (data not shown). We could speculate that not all cells in the lung respond to MnTBAP in the same way and therefore changes in MAP1LC3B could be masked.

## 3. Discussion

In the present study, we demonstrated for the first time that MnTBAP can increase BMPR2 levels in PAECs and pulmonary MVECs and MVSMCs isolated from iPAH patients, partly by inhibiting autophagy. Moreover, our experimental data in vivo indicate that MnTBAP treatment may be a promising therapy to reverse pulmonary vascular remodeling and benefit the RV.

PAH is a life-threatening disease for which no curative treatment is available. Since reduced BMPR2 is associated with PAH pathophysiology, restoring BMPR2 levels seems to be a suitable and promising alternative to effectively treat PAH patients. Indeed, increasing BMPR2 levels via adenoviral gene transfer into ECs of lungs was found to reverse hypoxic induced PH in rats [38]. In the present study, we demonstrated for the first time that MnTBAP increases BMPR2 levels in PAECs and pulmonary MVECs and MVSMCs isolated from iPAH patients. We have shown that even in the presence of CHX, MnTBAP increases BMPR2 levels in PAECs. This goes in line with our previous findings, where pre-treatment of HUVECs with MnTBAP for 16 h prior to addition of CHX increased BMPR2 stability [25]. In summary, our data suggest that MnTBAP modulates BMPR2 degradation and underlines the role of such compound as a favorable PAH treatment.

We recently demonstrated that BMPR2 is degraded through the lysosomal pathway in an autophagy-related fashion in PAECs [12], therefore we hypothesized that MnTBAP influences BMPR2 levels possibly through the regulation of autophagy. As expected, our results suggest that MnTBAP impairs functional autophagy in vitro; the exact mechanism of how MnTBAP works has yet to be elucidated. Furthermore, the possibility that MnTBAP increases BMPR2 levels by decreasing inflammation cannot be excluded, as inflammation has been shown to reduce BMPR2 levels [13,15] and to trigger autophagy [39,40].

As a cell-permeable superoxide dismutase mimetic, MnTBAP has been found to show beneficial effects in multiple disease models related to oxidative damages, including bleomycin-induced pulmonary fibrosis [27], carrageenan-induced pleurisy [28], lung contusion [26], renal fibrosis [29] and renal injury [24,30]. More interestingly, a previous study revealed that MnTBAP treatment regressed PH in Fawn hooded rats (FHR) by acting on PAMSCs [41]. Considering the previous and based on the positive findings of MnTBAP on BMPR2 in vitro, SuHx-induced PAH rats were treated with MnTBAP. Unlike FHR which spontaneously develops PH characterized by medial hypertrophy of the vessels [42], SuHx-induced PAH rats has been shown to be a reliable animal model replicating the characteristics of PAH, including occlusions of small-to-mild-sized pulmonary vasculatures, and the virtual unresponsiveness to current PAH treatments [43]. By pressure–volume loop and echocardiography analysis, we found that MnTBAP partly reverse established PAH in SuHx rats, as shown by reduced Ea and TPR. Further analysis by histology on the lungs revealed that MnTBAP reversed pulmonary vascular remodeling, in both the intima and media layers. This is consistent with the increase of BMPR2 levels after ECs and SMCs are treated with MnTBAP in vitro. Moreover, our finding on reduced media layer thickness is consist with a previous study, which showed that MnTBAP reduced media layer thickness in FHR by reducing hypoxia-inducible factor-1α and restoring voltage-gated potassium channels in FHR PAMSCs [41]. Based on the effects of MnTBAP on BMPR2, autophagy, oxidative stress and inflammation, as well as the interactions between these stimuli, MnTBAP treatment may reverse pulmonary vascular remodeling via multiple mechanisms.

In addition, it has to be noted that the survival of PAH patients is determined by the RV function [44,45]. Therefore, it is crucial that the compounds used to treat pulmonary vasculatures are beneficial or at least non-toxic to the heart. In a congestive heart failure model, MnTBAP treatment significantly ameliorated the symptoms by reducing oxidative stress [46]. However, the effects of MnTBAP on the RV have never been studied before. Importantly, we found for the first time that MnTBAP treatment is beneficial to the RV as shown by reduced RV stiffness, improved SV and TAPSE, and reversed RV dilation. The observed effects on the RV could be mediated by the anti-oxidation and anti-inflammation effect of MnTBAP, as oxidative stress and inflammation are largely increased in SuHx RV [37,47]. Or the beneficial effect on the RV are indirectly due to reduction of the RV afterload.

However, this study is limited by including only one animal model, thus it is unclear whether MnTBAP has direct beneficial effects on the RV, or benefits the RV indirectly due to reduction of the RV afterload. Besides the anti-oxidation and anti-inflammation effects, MnTBAP may also benefit the RV by modulating BMPR2 levels, particularly in ECs of the RV. Further studies using, e.g., pulmonary artery banding models could help to better understand the role MnTBAP on the RV. Moreover, it has to be noted that the observed effects of MnTBAP in vivo might be partly independent from the observed effects in vitro. Further studies to check BMPR2 and LC3B-II levels in PAEC and PASMCs from MnTBAP treated rats would be helpful to better understand the observed effects in the pulmonary vasculatures in vivo. Besides, a group of healthy control rats with MnTBAP treatment would be helpful to better understand the role of MnTBAP on healthy ECs and PASMCs. Although MnTBAP has not been approved to be used in humans, yet, we did not observe any adverse effect by MnTBAP in SuHx rats.

## 4. Materials and Methods

### 4.1. Antibodies and Reagents

The following anti-human antibodies were used: mouse anti-BMPR2 (612292, recognizing the cytoplasmic domain) from BD Biosciences (Vianen, The Netherlands), mouse anti-MAP1LC3B (5F10, 0231-100, recognizing the N terminus of MAP1LC3B) from Nanotools (Teningen, Germany) and mouse anti-SQSTM1 (sc-28359, recognizing the C terminus of SQSTM1) from Santa Cruz (Heidelberg, Germany). Rabbit anti-α/β-Tubulin (#2148) was from cell signaling (Leiden, The Netherlands) and mouse anti-glyceraldehyde-3-phosphate dehydrogenase (GAPDH) (MAB374) was from Millipore (Amsterdam-Zuidoost, The Netherlands). Rabbit (W4011) and mouse (W4021) horseradish peroxidase conjugated secondary antibodies were from Promega (Leiden, The Netherlands). Bafilomycin A_1_ (BafA1; B1793), cycloheximide (CHX; 01810) were obtained from Sigma Aldrich (St. Louis, MO, USA). Manganese (III) tetrakis (4-benzoic acid) porphyrin (MnTBAP; 475870) was from Merck (Calbiochem). Tumor necrosis factor alpha (TNF-α; 300-01A) was obtained from PeproTech (London, UK).

### 4.2. Cell Culture

Human pulmonary artery endothelial cells (PAECs) were purchased from Lonza (CC-2530, Basel, Switzerland) and were used between passages 4 and 8. Cells were maintained in Endothelial Cell Basal Medium 2 (Promo Cell, Heidelberg, Germany, C-22211), supplemented with ECGM-2 SupplementPack (Promo Cell, Heidelberg, Germany, C-39211) containing 0.02 mL/mL FCS, 5 ng/mL hEGF, 0, 2 g/mL hydrocortisone, 10 ng/mL hbFGF, 20 ng/mL insulin-like growth factor (R^3^-IGF-1), 0.5 ng/mL VEGF, 1 µg/mL Ascorbic Acid and 22.5 µg/mL heparin.

Primary human pulmonary microvascular endothelial and smooth muscle cells (MVECs and MVSMCs) were obtained from end-stage PAH patients and healthy tissues of lobectomy donors, as described before [48]. The tissue harvest and MVEC/MVSMC isolations were approved by the local ethics committees at the Amsterdam UMC (Amsterdam, The Netherlands) and written informed consent was obtained. MVECs were cultured in complete ECM medium supplemented with 1% pen/strep, 1% endothelial cell growth supplement, and 5% FCS (ScienceCell, Uden, The Netherlands). MVSMC were cultured in Dulbecco’s Modified Eagle Medium: F12 (Lonza, Basel, Switzerland, BE04-687F/U1) supplemented with 1% pen/strep and 10% FBS (ScienceCell, Uden, The Netherlands). Cell lines were routinely tested for mycoplasma contamination and only used if negative.

### 4.3. Western Blot Analysis

Western blot analysis was performed using standard techniques. In brief, cells were lysed in Laemmli buffer (0.12 M Tris HCl, pH 6.8, 4% SDS, 20% glycerol, 35 mM β-mercaptoethanol) and boiled for 5 min. Protein concentrations were measured using DC protein assay (Bio-Rad, 5000116). Equal amounts of total lysate were analyzed by sodium dodecyl sulfate polyacrylamide gel electrophoresis (SDS-PAGE). Proteins were transferred to polyvinylidene difluoride membrane (Millipore, Amsterdam-Zuidoost, The Netherlands, IPFL00010). Membranes were blocked with 5% non-fatty milk and incubated with the appropriate antibodies according to the manufacturer’s instructions. Membranes were then washed, incubated with appropriate peroxidase-conjugated secondary antibodies and developed by ECL (Bio-Rad, 1705061).

### 4.4. Quantitative Real-Time PCR

Total RNA was isolated using a NucleoSpin RNA kit (Macherey Nagel, Duren, Germany, 740955) according to manufacturer’s protocol. Total RNA was isolated using a NucleoSpin RNA kit (Macherey Nagel, Duren, Germany, 740955) according to manufacturer’s protocol. RNA was reverse transcribed using RevertAid RT Reverse Transcription Kit (ThermoFisher Scientific, Landmeer, The Netherlands, K1691). Generated cDNA was amplified with primer pairs for the indicated gene, using the CFX Connect Real-Time PCR Detection System (Bio-Rad). GAPDH was used as housekeeping gene. Quantification was performed relative to the levels of the GAPDH and normalised to control conditions. The data analysis was performed using the 2^−ΔΔCt^ method1. Primer sequences: *BMPR2* forward AACTGTTGGAGCTGATTGGC reverse CGGTTTGCAAAGGAAAACAC. *GAPDH* forward AGCCACATCGCTCAGACAC reverse GCCCAATACGACCAAATCC.

### 4.5. Flow Cytometry Analysis

PAECs were treated with HCQ (20 µM) and MnTBAP (50 µM). Cells were then trypsinized and incubated in Endothelial Cell Basal Medium 2 (Promo Cell, Heidelberg, Germany, C-22211), supplemented with ECGM-2 SupplementPack (Promo Cell, Heidelberg, Germany, C-39211) with Cyto-ID Autophagy Detection dye (Enzo Life Sciences, Brussels, Belgium, ENZ-51031-0050) at a dilution of 1:500 for 25 min at 37 °C. Subsequently, cells were washed and analyzed by flow cytometry. All data were analyzed using FlowJo software.

### 4.6. Experimental Pulmonary Arterial Hypertension

All experiments with animals were approved by an independent local animal ethic committee at Amsterdam UMC (Amsterdam, The Netherlands, study number 129-RUG18-02), and were carried out in compliance with guidelines issued by the Dutch government.

### 4.7. SuHx Rat Model of Pulmonary Hypertension and MnTBAP Treatment

Male Sprague-Dawley rats (*n* = 24; 170–210 g; Charles River, Sulzfeld, Germany) were used throughout the experiment. Rats were housed in standard conditions and food and water was available ad libitum. SU5416+Hypoxia (SuHx)-mediated PAH protocol was induced as described previously [37,49]. Briefly, rats were subjected a single injection of SU5416 ((25 mg/kg, 3037, Tocris Bioscience, Bristol, UK) followed by a 4-week transient exposure to 10% hypoxia (Biospherix Ltd., New York, NY, USA) maintained by a nitrogen generator (Avilo, Dirksland, The Netherlands) and re-exposed to normoxia for 6 weeks. Animals were randomized at week 6 receiving either saline (SuHx) or MnTBAP 10 mg/kg (MnTBAP) by intraperitoneal injection 3 times per week for 4 weeks. This dose and frequency of MnTBAP administration were according to previous reports in mice [29]. Due to loss of rats, hemodynamic data was collected from 6 rats in SuHx group and 7 rats in MnTBAP group. As control, data of healthy male Sprague-Dawley rats were taken from a previous published study of our group as (ctrl, *n* = 7) [37].

### 4.8. Hemodynamic Measurements

Echocardiography (Prosound SSD-4000 and UST-5542; Aloka, Tokyo, Japan) was performed at the beginning and the end of the treatment to measure the cardiac function, including SV, heart rate, PAAT, cl, TAPSE, RV wall thickness and RVEDD. PAAT/cl% was used to estimate RVSP, and TPR was calculated as mean pulmonary artery pressure/cardiac output.

At the end of the experiment, rats were anaesthetized with 4% isoflurane for hemodynamic assessment via open-chest RV catheterization (Millar Instruments, Houston, TX, USA). RVSP was determined from steady state measurement, as well as Ea. Pressure–volume loops after vena-cava occlusion were obtained and used to measure Ees and Eed. RV–arterial coupling was calculated as Ees/Ea. One out of 13 rats with hemodynamic measurement was detected to have much higher Ea than the rest. Under the circumstance that Ea is normally distributed in historical database, the Dixon outlier test was applied and it revealed that this rat—in contrast to all other individuals of the MnTBAP treated group—was characterized as a significant outlier. Based on a previous literature [50], this rat was excluded from all statistical evaluations. Analysis of echocardiography and pressure–volume loops was done blinded.

### 4.9. Morphometry of the Pulmonary Vasculatures

After RV catheterization, animals were exsanguinated. Lungs were collected and inflated with an 1% solution of low-melt-agarose and fixed in formalin and embedded in paraffin. To determine the pulmonary vascular remodeling, paraffin embedded 5-µm-thick lung sections were stained with Elastica van Giesson. Small pulmonary arteries were divided into three classes, based on external diameters: 0–30, 30–60 and 60–100 μm [49]. Occluded pulmonary arteries with an external diameter between 0–100 μm were counted. Media and intima wall thickness of pulmonary vasculatures classified by the external diameters, were measured and recorded separately as described previously [49], and as described in Appendix A. Morphometry analysis of the pulmonary vasculatures was done blinded.

#### 4.9.1. Immunofluorescence Staining

Rat lungs were collected and inflated with an 1% solution of low-melt-agarose and fixed in formalin and embedded in paraffin. Five micrometer-thick lung paraffin sections were deparaffinized and rehydrated. Sections were boiled for 40 min in Vector^®^ Antigen Unmasking Solution (Vector, Burlingame, USA) using a pressure cooker. After blocking with goat serum 10% (ThermoFisher Scientific, Massachusetts, USA), sections were incubated overnight at 4 °C with primary antibodies directed against 8-OHdG (1:150; bs-1278R, Bioss Antibodies, Woburn, MA, USA), co-stained with von Willebrand factor (1:1000; ab8822, Abcam, Cambridge, UK), and alpha smooth muscle actin (1:1000; Sigma-Aldrich, St. Louis, MO, USA). All sections were mounted with ProLong^®^ Gold antifade reagent (Invitrogen, Massachusetts, USA) containing DAPI.

#### 4.9.2. Immunofluorescence Quantification

Images were acquired on a Marianas digital imaging microscopy workstation (Intelligent Imaging Innovations (3i), Denver, CO, USA). SlideBook imaging analysis software (SlideBook 6, 3i) was used to semi-automatically quantify the images. Pulmonary vascular 8-OHdG mean relative fluorescence intensity was semi-automatically quantified and measured over twenty vessels.

### 4.10. Statistical Analysis

Statistical analysis was performed using Prism for Windows (GraphPad 8 software). Normality of data was checked with Kolmogorov-Smirnov test and either log-transformation or non-parametric test was performed if data was not normally distributed. Unpaired two-sided student’s test was used to calculate statistical differences between two groups. The survival estimates were performed by log rank (Mantel-Cox) test between SuHx rats with/without MnTBAP treatment. Multiple comparisons were assessed by one-way ANOVA, followed by Bonferroni’s post-hoc test. Two-way ANOVA for repeated measurements followed by Bonferroni post-hoc test was used to compare parameters collected from echocardiography. A *p*-value of <0.05 was considered statistically significant. Data presented as mean ± SD.

## 5. Conclusions

We demonstrated for the first time that MnTBAP modulates BMPR2 degradation in ECs and SMCs of PAH patients, partly by inhibiting autophagy. Moreover, our in vivo data revealed that MnTBAP treatment can partly reverse RV afterload and pulmonary vascular remodeling in established experimental PAH. Importantly, it is beneficial to the RV by reducing RV afterload. Collectively, MnTBAP may be a promising intervention for PAH.

## Figures and Tables

**Figure 1 ijms-21-04130-f001:**
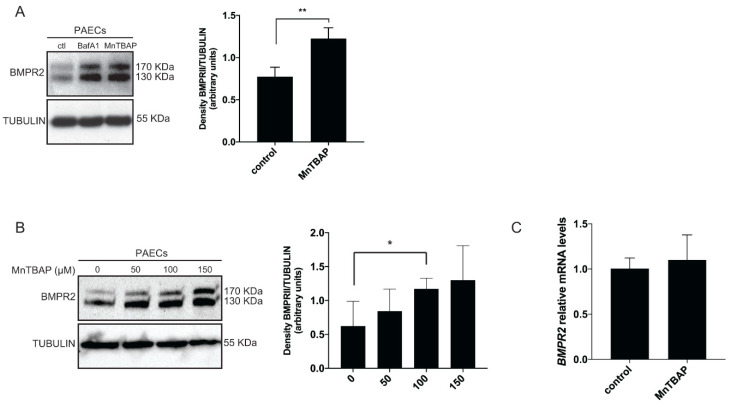
MnTBAP increases BMPR2 at the post-transcriptional level. (**A**) pulmonary artery endothelial cells (PAECs) were treated with MnTBAP (50 µM) and BafA1 (20 nM) for 16 h. Left panel: BMPR2 protein expression was analyzed by western blot. BMPR2 protein levels increased after treatment. Tubulin is used as a loading control. Representative results of at least 3 independent experiments are shown. Right panel: Quantification of BMPR2 protein levels normalized for tubulin. (**B**) PAECs were treated with 50 µM, 100 µM and 150 µM of MnTBAP for 16 h. BMPR2 levels increased in a dose dependent manner. (**C**) *BMPR2* mRNA expression analyzed by qRT-PCR remains constant after PAECs were treated with MnTBAP (50 µM). Data presented as mean ± SD. * *p* < 0.05, ** *p* < 0.01.

**Figure 2 ijms-21-04130-f002:**
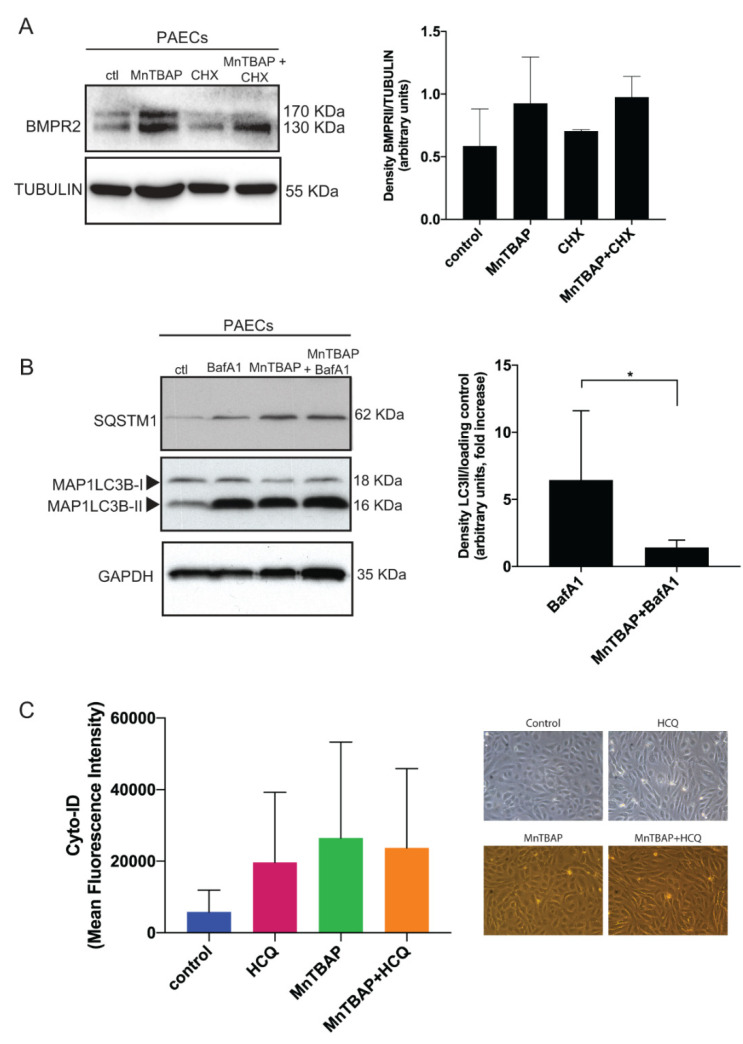
MnTBAP inhibits BMPR2 degradation by blocking autophagy. (**A**) MnTBAP restores inhibitory effects of cyclohexamide (CHX) on BMPR2 protein expression. PAECs were treated with MnTBAP (50 µM) for 16 h, cycloheximide for 2 h and the combination. BMPR2 protein expression was analyzed by western blot. Tubulin is used as a loading control. Representative results of at least 3 independent experiments are shown. (**B**) PAECs were treated with MnTBAP (50 µM) in the presence or absence of BafA1 (20 nM) for 16 h and lysed directly after the treatment. Left panel: MAP1LC3B-II protein levels were analyzed by western blot. Glyceraldehyde-3-phosphate dehydrogenase (GAPDH) is used as a loading control. Representative results of at least 3 independent experiments are shown. Right panel: Western blot quantification of MAP1LC3B-II normalized for the loading control. The data are presented as fold increases relative to cells treated with BafA1. (**C**) Left panel: Flow cytometry-based analysis of the quantification of autophagic vesicle content in PAECs by means of the Cyto-ID dye. Cells treated with MnTBAP (50 µM) for 16 h with and without hydroxychloroquine (HCQ) (20 µM) and analyzed directly after the treatment. Right panel: Representative light microscopy images of PAECs treated for 16 h with MnTBAP (50 µM), HCQ (20 µM) and the combination, to show the status of the cells after MnTBAP treatment. Upper panels, bright field images; lower panels, fluorescent images. Magnification: ×20. Data presented as mean ± SD. * *p* < 0.05.

**Figure 3 ijms-21-04130-f003:**
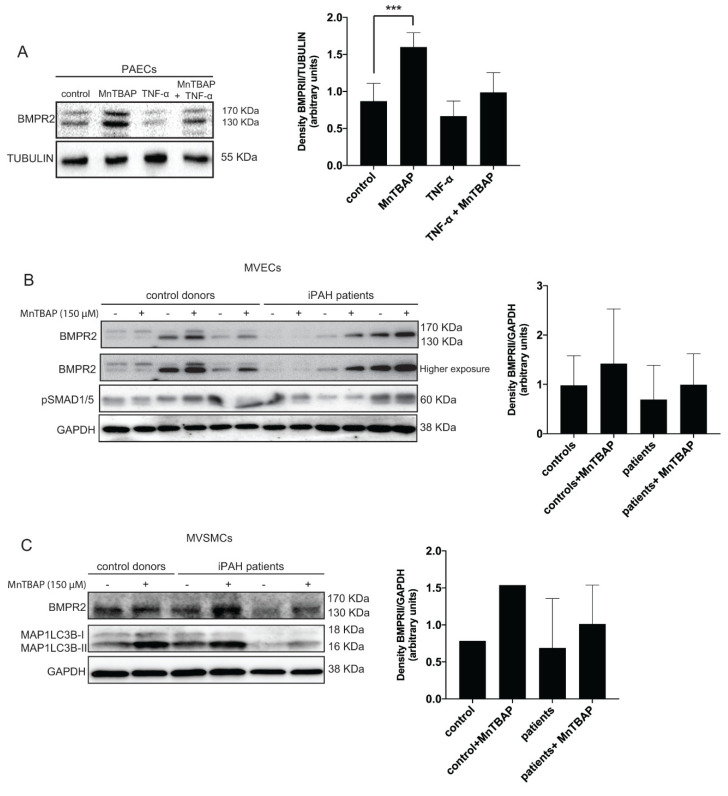
MnTBAP restores the levels of BMPR2 in pulmonary microvascular endothelial cells (MVECs) and microvascular smooth muscle cells (MVSMCs) isolated from idiopathic pulmonary arterial hypertension (iPAH) patients. (**A**) hPAECs were treated with MnTBAP (150 µM) for 16 h and TNF-α (10 ng/mL) for 2 h. Left panel: BMPR2 protein expression was analyzed by western blot. Tubulin is used as a loading control. Representative results of at least 3 independent experiments are shown. Right panel: Quantification of BMPR2 protein levels. (**B** and **C**) Pulmonary MVECs and MVSMCs isolated from iPAH patients were treated with MnTBAP (150 µM) for 24 h. Left panel: BMPR2, pSMAD1/5 and MAP1LC3B-II protein expression was analyzed by western blot. GAPDH is used as a loading control. Representative results of at least 3 independent experiments are shown. Right panel: Quantification of BMPR2 protein levels. Data presented as mean ± SD. *** *p* < 0.001.

**Figure 4 ijms-21-04130-f004:**
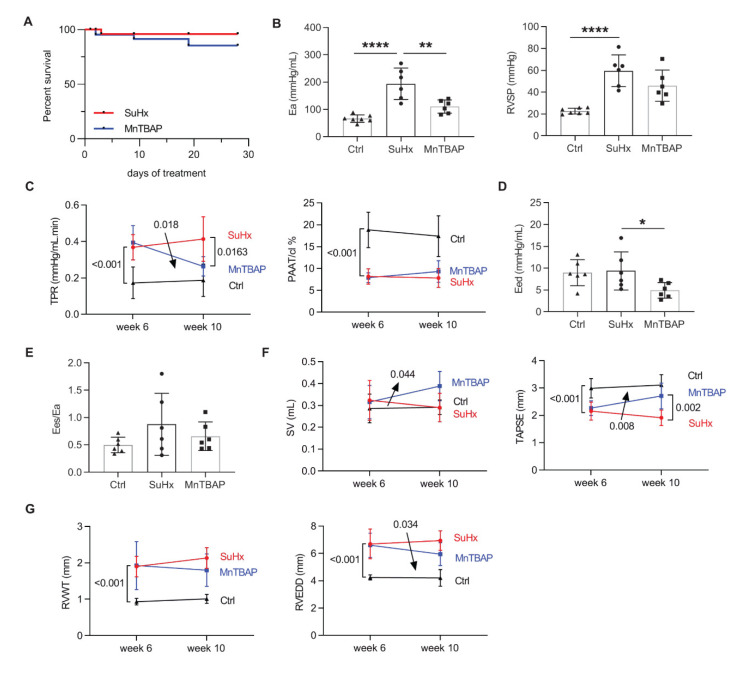
MnTBAP reduces right ventricular (RV) afterload and improves cardiac function in SuHx-induced PAH in rats. (**A**) MnTBAP treatment did not affect survival in PAH rats. (**B**) Pressure–volume loop analysis shows that MnTBAP treatment reduced RV afterload (Ea) at week 10, with remained RVSP. (**C**) Echocardiography analysis shows that MnTBAP delayed the progression of TPR from week 6 to week 10, and reduced TPR at week 10, while PAAT/cl% was not affected. (**D**,**E**) Pressure–volume loop analysis shows that MnTBAP reduced RV stiffness (Eed) at week 10, with remained RV-arterial coupling (Ees/Ea). (**F**,**G**) MnTBAP treatment delayed progression towards RV failure as shown by echocardiography at week 6 and week 10, with decreased RVEDD, and improved SV and TAPSE. Arrows represent significant interaction of the two-way ANOVA. Data presented as mean ± SD. * *p* < 0.05, ** *p* < 0.01, **** *p* < 0.0001 versus SuHx. Ea = arterial elastance, RVSP = right ventricular systolic pressure, TPR = total pulmonary resistance, PAAT = pulmonary artery acceleration time, cl = cycle length, Eed = end diastolic elastance, Ees = end systolic elastance, SV = stroke volume, TAPSE = tricuspid annular plane systolic excursion, RVWT = right ventricular wall thickness, RVEDD = right ventricular end diastolic diameter. MnTBAP reduced pulmonary vascular remodeling and oxidative stress in experimental PAH.

**Figure 5 ijms-21-04130-f005:**
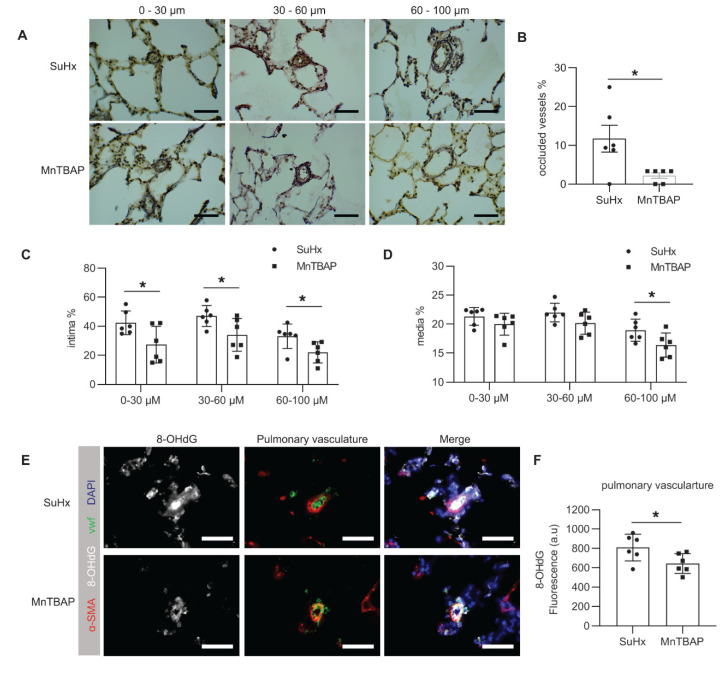
MnTBAP reduces pulmonary vascular remodeling and oxidative damage in SuHx-induced PAH in rats. (**A**) Representative images of the pulmonary vasculatures by Elastica van Giesson staining. Scale bar: 50 μm. (**B**) Quantification of occluded vessels in SuHx and MnTBAP-treated rats. (**C**) Quantification of intima remodeling ratio in pulmonary vasculatures between 0–30, 30–60 and 60–100 μm, respectively. (**D**) Quantification of media remodeling ratio in pulmonary vasculatures between 0–30, 30–60 and 60–100 μm, respectively. (**E**) Representative images of 8-OHdG immunofluorescence staining in the lungs. Scale bar: 50 μm. α-SM (red), vwF (green) and DAPI (blue) were co-stained with 8-OHdG (white). (**F**) Quantification of 8-OHdG intensity shows that MnTBAP reduced 8-OHdG within the pulmonary vasculatures. * *p* < 0.05. Data presented as mean ± SD. 8-OHdG: 8-Oxo-2′-deoxyguanosine, α-SM: alpha smooth muscle actin, vwF: von Willebrand factor.

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
