# Peer review of "MnTBAP Reverses Pulmonary Vascular Remodeling and Improves Cardiac Function in Experimentally Induced Pulmonary Arterial Hypertension"

_ijms, 2020, doi:10.3390/ijms21114130_

Round 1

Reviewer 1 Report

In this MS, Gomez-Puerto et al have investigated the participation of BMPR2 signals in the context of autophagy in improving the PAH symptoms in the Su-Hyx animal model. Authors have also utilized invitro tools and performed molecular characterization, which is the strength and merit of the MS, while the lack of control for the invivo studies dampens the enthusiasm. Here are the comments -

  1. Across the MS everywhere (wherever Western data are displayed), authors are required to include the densitometry quantification data for all westerns - Fig 1A, 1B, 2A, 2B, 2C and 3.
  2. What is the significance of cell images in Fig 2C? Can authors show the cells on higher magnification?
  3. In order to study the muscularization of pulmonary vessels, authors are required to demonstrate the distribution of muscularized vessels based on the sizes - For instance, 0-25 uM, 25-50uM and 50-100 uM.
  4. Scale is missing in Sup.Fig 2 and 3
  5. What is the rationale for the invivo dose of MnTBAP?
  6. Are morphometric analysis are blinded?
  7. Prior testing the effect of MnTBAP in PAH ECs or VSMCs, authors have investigated the same in PAEC, which suggest that MnTBAP does have a role even in the wild type cells. In this context, the invivo experiment is missing the appropriate controls. Authors are required to include the control animals in presence and absence of MnTBAP and recommended to demonstrate all the experimental outcome discussed in the Fig 4. After the inclusion of data, authors are required to reanalyze the data, by 2-way anova.
  8. Knowing the fact that MnTBAP has a role in containing ox. stress and also Ox. Stress has a role in the pathogenesis of PAH, checking a couple of molecular markers of ROS species (western or PCR) would provide more molecular insights and bolster the storyline.
  9. As a proof of concept, authors are required to demonstrate autophagy by western, supplementing the existing IHC data in Sup.Fig 3.

Reviewer 2 Report

The manuscript seems to cover an interesting topic as it evaluates the potential efficacy of MnTBAP that could reverse pulmonary vascular remodeling and improve cardiac function in experimental PH. The presented data are wide, and methodology appears appreciate. Minor comments/questions can include as follows:

  • The authors did not provide the results for negative control group (Sham), why?
  • The amount of dispersion of a set of data values should be presented using SD (standard deviation) parameter instead of SEM. SEM, as a measure of precision for an estimated population mean, is not a descriptive statistics and should not be used as such.
  • If any procedures were blinded?
  • Please complete the test for normality assessments (statistics) and more detailed results for two-way ANOVA
  • Please complete the description (Figure 4a) – that the measurement concerned 10th week of the study
  • What are the limitations of the study
  • What was the rationale for such a schedule (ip injection of MnTBAP 3 times a week)?
  • The survival analysis would be valuable addition to current data

Reviewer 3 Report

This manuscript describes the work by Gomez-Puerto et al., in which the authors investigate the effect of MnTBAP in Su-Hx model of PAH. The authors show that treatment with MnTBAP reverses pulmonary vascular remodeling and improves cardiac function through a mechanism involving autophagic regulation of BMPR2. While the manuscript is well written, there are many concerns which limit the interpretation of the data and make it difficult to derive convincing conclusions as detailed below.

Major comments:

  1. Representative data in Figure 2A: While MnTBAP treatment increases BMPR2 expression in the presence of protein synthesis inhibitor CHX compared to CHX alone, BMPR2 expression appears to be decreased compared to MnTBAP alone. This may suggest the involvement of protein translation, in addition to autophagic regulation, in modulating BMPR2 expression in response to MnTBAP. The authors are encourged to provide western blots quantification of BMPR2 in the same experimental setting to help to clarify this concern.

  1. Figure 2B: While SQSTM1 and LC3B-II levels are similar in PAECs treated with MnTBAP and BafA1, alone or in combination, in the representative western blots, the quantification data indicate a 3-fold difference between BafA1 alone and the combination group. Also, GAPDH was mentioned as loading control for Figure 2B, however, quantification was made using tubulin. The authors are recommended to clarify the corresponded results.

  1. Quantification data is also needed for Figure 3A.

  1. The observed changes in in vivo pulmonary vasculature and the in vitro cell culture may be independent. The authors are suggested to check BMPR2 and LC3B-II levels in PAEC and/or PASMCs from MnTBAP-treated rats.

  1. It is not clear whether MnTBAP has a direct effect in modulating BMPR2 levels in the RV, particularly endothelial cells of RV, for the improved RV function. The authors are recommended to further assess that.

  1. The authors mentioned about the loss of animals, 6 and 5 rats for control and MnTBAP groups, respectively, during the experimental period. Is this related to improved survival of the treatment? The authors are recommended to provide more detailed information.

Minor comments:

  1. In the text, the authors mention that MnTBAP increases BMPR2 levels in the presence of CHX in PAECs, as well as in MVECs and MVSMCs from iPAH patients. However, the authors do not provide any data from MVECs and MVSMCs.

  1. Fonts are different in Figure4 legend.

Round 2

Reviewer 1 Report

Authors have met all suggestions and no more corrections required

Author Response

We thank the reviewer for all the comments and suggestions!

Reviewer 3 Report

The reviewer has no further comment.

Author Response

(The authors gave the same response as above.)
